# Classification of Central Venous Catheter Tip Position on Chest X-ray Using Artificial Intelligence

**DOI:** 10.3390/jpm12101637

**Published:** 2022-10-03

**Authors:** Seungkyo Jung, Jaehoon Oh, Jongbin Ryu, Jihoon Kim, Juncheol Lee, Yongil Cho, Myeong Seong Yoon, Ji Young Jeong

**Affiliations:** 1Department of Emergency Medicine, College of Medicine, Hanyang University, Seoul 04763, Korea; 2HY-Medical Image and Data Artificial Intelligence System (MIDAS) LAB, Hanyang University, Seoul 133791, Korea; 3Department of Software and Computer Engineering, Ajou University, Suwon 11759, Gyeonggi-do, Korea

**Keywords:** image, central venous catheter, deep learning, machine learning, artificial intelligence, AI

## Abstract

Recent studies utilizing deep convolutional neural networks (CNN) have described the central venous catheter (CVC) on chest radiography images. However, there have been no studies for the classification of the CVC tip position with a definite criterion on the chest radiograph. This study aimed to develop an algorithm for the automatic classification of proper depth with the application of automatic segmentation of the trachea and the CVC on chest radiographs using a deep CNN. This was a retrospective study that used plain chest supine anteroposterior radiographs. The trachea and CVC were segmented on images and three labels (shallow, proper, and deep position) were assigned based on the vertical distance between the tracheal carina and CVC tip. We used a two-stage approach model for the automatic segmentation of the trachea and CVC with U-net^++^ and automatic classification of CVC placement with EfficientNet B4. The primary outcome was a successful three-label classification through five-fold validations with segmented images and a test with segmentation-free images. Of a total of 808 images, 207 images were manually segmented and the overall accuracy of the five-fold validation for the classification of three-class labels (mean (SD)) of five-fold validation was 0.76 (0.03). In the test for classification with 601 segmentation-free images, the average accuracy, precision, recall, and F1-score were 0.82, 0.73, 0.73, and 0.73, respectively. We achieved the highest accuracy value of 0.91 in the shallow position label, while the highest F1-score was 0.82 in the deep position label. A deep CNN can achieve a comparative performance in the classification of the CVC position based on the distance from the carina to the CVC tip as well as automatic segmentation of the trachea and CVC on plain chest radiographs.

## 1. Introduction

In the emergency room and intensive care unit, temporary central venous catheters (CVC) placed on internal jugular, subclavian, and femoral vein catheters are frequently used for drug administration, central venous pressure monitoring, and short-term hemodialysis [1,2]. The ideal position of the CVC tip via the internal jugular vein (IJV) and subclavian vein (SCV) is in the bottom one-third of the superior vena cava (SVC) near the junction of the right atrium (RA) and SVC [3,4,5]. If the CVC is implanted too deeply in the RA, central venous pressure monitoring may still be performed, but malignant arrhythmias and catheter erosion may develop, leading to perforation, hemothorax, and cardiac tamponade [6,7]. If it is positioned too shallowly, the risk of local venous thrombosis, catheter malfunction, and cranial retrograde injection increases [8,9].

Clinicians can attempt to position the CVC tip for optimal placement using a variety of techniques during and after the procedure. The intracavitary electrocardiographic method is currently recommended in international guidelines for determining the proper depth of the CVC tip [10,11]. Transesophageal echocardiography is the gold standard for assessing the placement of the CVC tip [12]. Transthoracic echocardiography with a contrast enhancement may be an accurate method of detecting CVC misplacements following IJV cannulation [13]. However, these techniques may not be readily accessible in clinical practice settings such as the emergency room and intensive care unit. Postprocedure chest radiograph imaging was not necessary because a chest radiograph is not a reliable test to confirm the tip location, and it has the major disadvantage of being post-procedural [14]. However, confirmation of tip location by this radiograph remains acceptable practice and is required in the absence of technology used during the procedure [14,15,16]. Because of its widespread availability and low cost, it might still be the most common tool for confirming the position of the CVC tip [16,17]. 

Recent studies utilizing deep convolutional neural networks (CNNs) have described the CVC on chest radiography images [18,19,20,21,22,23]. Among them, two studies showed that the network for segmentation can increase the effectiveness of CVC tip detection and type classification [18,19]. Only one study demonstrated that deep learning using the National Institute of Health (NIH) ChestXRay14 dataset could classify the catheter malposition (normal, abnormal, and borderline) [21]. However, there was no definite criterion for the placement of the CVC tip on the chest radiograph and its performance. The vertical distance between the tracheal carina and the CVC tip in CXR could be a simple and precise method to confirm not only the safe placement of the CVC tip but also its optimal positioning for accurate hemodynamic monitoring. [16] This study aims to develop an algorithm for the automatic classification of the proper depth based on the vertical distance between the tracheal carina and the CVC tip (shallow, proper, and deep position) with the application of automatic segmentation of the trachea and the CVC on chest radiographs using a deep CNN.

## 2. Methods

### 2.1. Study Design

This was a retrospective study employing plain chest supine anteroposterior (AP) radiographs for automatic segmentation of the trachea and the CVC and classification of the distance between the carina of the trachea and the tip of the CVC using a deep CNN. The study was conducted between September 2021 and July 2022 at a regional emergency center at a tertiary hospital in Seoul (Seoul, Korea) using data from January 2017 to August 2021. The Institutional Review Boards (IRBs) at Hanyang University Hospital approved this study (reference number 2021-08-063) and waived the requirement for informed consent. All methodologies and procedures were conducted in line with the Helsinki Declaration.

### 2.2. Dataset of Participants

A flowchart of data collection and analysis is shown in Figure 1.

#### 2.2.1. Extraction and Categorization of Images on Chest Radiograph with CVC

We organized and collected supine AP chest radiographs of 18 to 80-year-old patients who underwent the CVC procedure in the emergency room between January 2017 and August 2021. Images with the CVC were extracted by searching the comment of “C line” or “C-line” using the picture archiving and communication system (PACS, Centricity, GE Healthcare, Milwaukee, WI, USA). Except for the permanent catheter and the temporary CVC implanted through the femoral vein, we included images of the temporary CVC inserted through the IJV and SCV. We omitted images with extensive pleural effusion, hemothorax, and pneumothorax, as well as low-quality images caused by serious trauma to anatomical structures. We did not, however, exclude images with an endotracheal tube, electrodes, or nasogastric tubes. In our emergency room, IJV and SCV catheters are inserted to a depth of 14 to 18 cm, depending on the direction and vein type, using almost real-time ultrasound guidance or infrequently landmarks [24,25]. Kang et al. reported that the distance of −6.7 to 15.6 mm on the chest radiograph is a simple and accurate guide for confirming the safe placement of the CVC tip and optimal positioning for accurate hemodynamic monitoring [15]. The area under the curve (AUC) was 0.987 for the SVC entrance of the CVC tip and 0.965 for the RA insertion of the CVC tip. With this range, we classified chest radiograph images under three labels (shallow, proper, and deep position). Two authors (Jung, S. and Oh, J.) agreed on each position of the carina and the CVC tip and drew two horizontal lines between them. We measured the vertical distance from the carina to tip (CTD) in millimeters (mm) using the PACS scale. These images were assigned one of three CVC position labels: (1) shallow CVC location (−6.7 mm), (2) proper CVC position (between −6.7 and 15.6 mm) (3) deep CVC position (more than 15.6 mm) based on the vertical distance between the carina and CVC tip [15]. When storing images for data collection, these were extracted and stored as digital imaging and communication in medicine (DICOM) format images utilizing a PACS system without any personal information. Before coding and saving, arbitrary numbers were allocated to photos before they were taken at random and stripped of any personally identifying information.

#### 2.2.2. Segmentation of the Trachea and the CVC on Images

In agreement with another author (Oh, J.), an author (Jung, S.) painted the entire trachea and the entire CVC area as binary masks to segment them. Using the AVIEW program, these segmented images were saved as NIfTI (Neuroimaging Informatics Technology Initiative) files (AVIEW-Research, Corelinesoft, Seoul, Korea).

### 2.3. Proposed Models for Classification of Three Classes of CVC Position with Application of Automatic Segmentation Using Deep CNN

Figure 2 represents the work flow of our classification model. It classifies the CVC position on chest radiograph images with three labels: shallow, proper, and deep position. Inspired by [21,26], we proposed a two-stage classification model that segments the trachea and CVC region first, then uses the segmentation results for the classification of the CVC position. To segment the trachea and the CVC regions, we used U-net^++^ (unet^++^) with fine-tuning of our own dataset. For the classification of the segmentation result, we utilized the EfficientNet-b4 (effnet) model, which required only a single segmented image as an input. We employed the initially pretrained EfficientNet on the ImageNet dataset and then fine-tuned its parameters using our own dataset. The input image and binary masks of the trachea and catheter identified by our segmentation network were concatenated and fed to the classification network. We resized input images as 224-by-224 resolution and trained them with 50 epochs and 32 batches. The AdamW optimizer and OneCycleLR scheduler were used to minimize the cross-entropy loss during training, and the learning rate and weight decay factor were set to 0.0001 and 0.005, respectively. All training and testing phases were carried out on a RTX 3090 GPU, and we implemented our model as PyTorch library (PyTorch).

### 2.4. Experiments

We separately trained the proposed models for the segmentation and classification tasks and evaluated their performance using five-fold cross-validation on a training and validation set comprised of segmented images. Images of chest radiographs were randomly divided into five sets; four of the five sets were used for training, and the remaining set was used for validation. This strategy would allow us to train the five models using five distinct training and validation datasets to validate the model generalization. Finally, we trained the optimized model with whole sets of training images and evaluated the classification performance of the CVC tip position using a test set.

### 2.5. Primary Outcomes

The primary outcome was a successful three-class problem based on the relationship between the tracheal carina and the tip of the CVC. We selected the prediction label with the highest prediction probability from the three-class labels, validated five-fold performance using a module with accuracy, precision, recall, and F1-score, and analyzed these values according to the performance of each label. The proportion of accurate predicted images relative to the total images of predictions was the accuracy. Precision was the ratio of correctly positive predicted images by our network to all positive predicted images, which was the positive predictive value. Recall was sensitivity or the proportion of correctly positive predicted images by our network to all real positive images. F1-score was the harmonic mean of precision and recall.

### 2.6. Statistical Analysis

The data were compiled using a common spreadsheet program (Excel 2016; Microsoft, Redmond, WA, USA) and analyzed using NCSS 12 (Statistical Software 2018, NCSS, LLC. Kaysville, UT, USA, http://www.ncss.com/software/ncss (accessed on 31 May 2022)). The Shapiro–Wilk test was performed to demonstrate the normal assumption of all datasets. Depending on the normality, we generated descriptive statistics and presented them as median and interquartile range (IQR) or mean and standard deviation (SD) for continuous data. Student’s tests or Mann–Whitney tests were used to compare the CTD between the segmented and segmentation-free data in same label. *p*-values < 0.05 were considered statistically significant.

## 3. Results

Out of the total 808 images, 207 were categorized as 77 images of shallow position, 80 images of proper position, and 50 images of deep position for training and five-fold validation. The distances between the tracheal carina and the CVC tip did not differ significantly between segmented and segment-free image datasets except in the proper position (Table 1). Table 2 represents the results of five-fold validation for the classification of the three-class problem based on the distance between the CVC tip and the carina. Overall accuracy (mean (SD)) of five-fold validation was 0.76 (0.03).

Table 3 shows the confusion matrix and overall performance and each label for classifying the segmentation-free images following training with all segmented images. The overall accuracy was 0.73. The average accuracy, precision, recall, and F1-score were 0.82, 0.73, 0.73, and 0.73, respectively. Accuracy had the highest value of 0.91 in the shallow position label, whereas the F1-score had the highest value of 0.82 in the deep position label.

## 4. Discussion

In this study, we first proposed a deep CNN for an automatic three-label classification model for the proper position of the CVC tip relative to the tracheal carina using chest radiograph images. In tests with segmentation-free images, the overall accuracy was 0.73 which represents the true positive proportion of all cases including the three classifications. In each label classification, the shallow label classification had the best accuracy at 0.91. This indicates that our network accurately classified 91 out of 100 images as shallow label images and non-shallow label images. The precision of the deep label classification was 0.76, the highest. This means that when our network classifies 100 images as deep label images, 76 images of the predictions are accurate, while the remaining 24 images are wrong. The most sensitive was the deep classification with 0.88 of recall. This indicates that 88 out of 100 deep images are accurately predicted by our network, while the remaining 12 are incorrectly classified. F1-score, the harmonic mean of precision and recall was the highest at 0.82 in the deep classification since precision and recall were both the highest. 

Five-stage research questions covering presence, detection of the tip, course, type, and satisfactory position were answered for the study about catheter positions on radiographs using a deep CNN [23]. Sabramanian et al. demonstrated that the automatic detection and type classification of the CVC on chest radiographs is feasible with a high performance using a modified U-net for segmentation and a random forest for type classification [22]. However, they did not conduct the classification of proper position of the CVC tip. Yu et al. reported that their novel multi-task deep CNN using U-net for catheter segmentation and Faster R-CNN/VGG-net for tip detection achieved the best performance with an F1-score of 0.74 for detecting the tip of the peripherally inserted central catheter (PICC) in chest radiograph images [18]. Lee et al. demonstrated that absolute distances from ground truth to predicted mask using PICC line segmentation and tip region of interest with full CNN were 3.10 mm on mean, with a standard deviation of 2.0 mm [19]. However, there was no classification report for the PICC tip position. Khan et al. reported that their proposed network achieved about 0.98 for macro-average AUC in 11 labels, including the CVC, nasogastric tube, and endotracheal tube placement [21]. There was no result for classification of only CVC position. In our test with U-net^++^ for CVC line segmentation and EfficientNet-b4 for the CVC position classification, overall accuracy for the three labels was 0.73. Average accuracy and F1-score were 0.82 and 0.73, respectively. However, we did not estimate the accuracy of tip detection and the difference between the ground truth and the predicted mask of the CVC tip. There is no deep learning study for the proper position of the CVC tip that compares to our network’s performance. In our previous study for the proper position of the endotracheal tube tip on chest radiography using deep CNN, we achieved high sensitivity and specificity about 0.85~0.93 for shallow and deep positioning of the endotracheal tube while the precision and F1-score were low, at about 0.32~0.46 for them [26]. If the postoperative chest radiograph is to be used as a screening tool to assess the optimal depth of the CVC tip, it should be more sensitive. The recall (sensitivity) of the shallow and appropriate images were 0.56 while the recall of the deep image was 0.88. We think that the performance of our network using the chest radiograph could be acceptable as a screening tool when we focus on fatal issues that arise in the deep cases of CVC tip rather than the shallow cases. We believe that chest radiography would still be useful for the conformation for the proper depth of the CVC tip when intracavitary electrocardiographic and ultrasound is not available. If our network for classification of the CVC tip on chest radiograph using deep CNN is developed as a suitable software, alarm messages from this software could aid physicians working in busy and human resources-limited environments such the emergency room.

In a study of deep CNNs and classification of 12 categories based on the distance from the carina and the endotracheal tube tip at 1.0 cm intervals, the performance for detecting shallow position images of the tube tip was 0.67 for recall, while for deep position images, it was 0.90. This was due to the network mislabeling the ground truth of 70–80 mm above the carina for a prediction of 60–70 mm, which reduced the sensitivity [27]. In this study, we classified three labels for the CVC position with one decimal point between −6.7 and 15.6 mm. The recall for detecting shallow CVC position was 0.56, while it was 0.88 for detecting deep CVC position. We believed this was difficult work in comparison to the study on endotracheal tube positioning using deep CNN and chest radiographs because of the narrow range of classification and the thin tip of the CVC catheter.

There were several limitations in this study. Regardless of the CVC insertion site, we applied the classification for the proper depth of the CVC tip based on the same criteria. It may actually be different according to right and left. Our proposed method could not detect the caudal position of the CVC because we excluded these images. We used data from chest radiographs taken at an emergency room from a single center, and this network model may not be applicable to other environments. The patient’s position influences chest radiographic verification of catheter tip position, and anatomical variation may also impact the interpretation of the tip position [28,29]. Finally, we did not compare our algorithm’s effectiveness as a screening tool to that of clinicians regarding aspects of inappropriate CVC tip position detection and time to detection. 

## 5. Conclusions

Deep CNNs can achieve a comparative performance in the classification of the CVC position based on the distance from the carina to the CVC tip as well as on automatic segmentation of the trachea and CVC on plain chest radiographs.

## Figures and Tables

**Figure 1 jpm-12-01637-f001:**
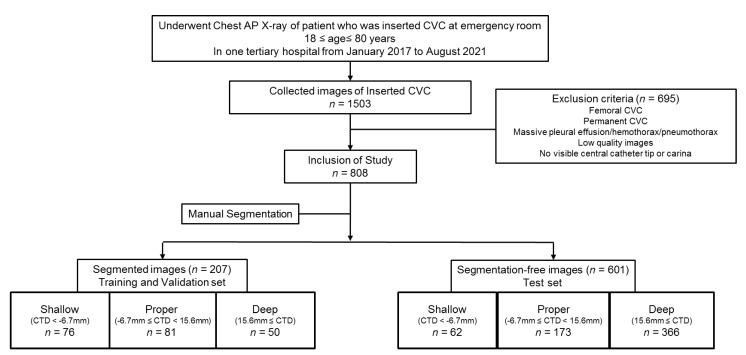
Flowchart of data collection for training/validation with segmented images and a test set with segmentation-free images in the study. All the images are classified according to CTD. CVC, central venous catheter; CTD, distance from the carina to tip of the catheter.

**Figure 2 jpm-12-01637-f002:**
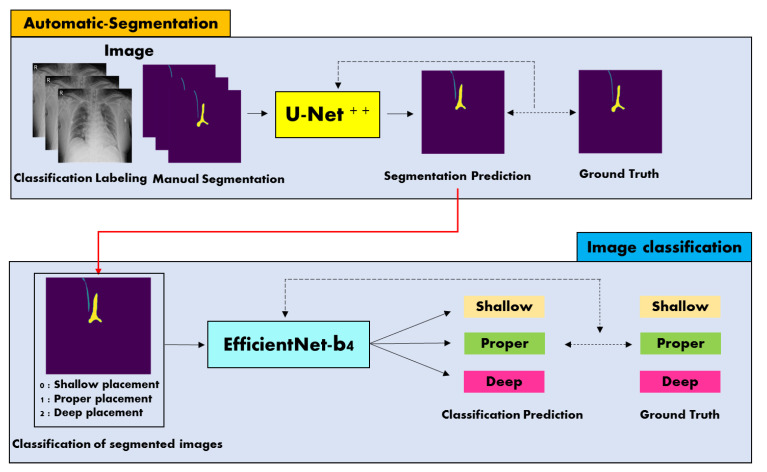
Work flow of our segmentation model of the trachea and CVC as well as three-class problem for correct position of the CVC tip on chest radiographs. U-net^++^ generates segmentation results as binary images. These binary segmentation images are fed into EfficientNet-b4 as an additional input to enhance the classification accuracy of the three-class problem of CVC position.

**Table 1 jpm-12-01637-t001:** Label-based comparison of the distances between the carina of the trachea and the tip of the CVC in segmented and segmentation-free image datasets.

		The Distance from the Carina to Tip of the CVC (CTD), Median (IQR)		
		Segmented Images for the Training and 5-Fold Validation	Segmentation-Free Images For the Test	*p*-Value
Labels	Shallow, mm	−16.39 (−26.04, −11.23)	−19.11 (−26.31, −12.46)	0.30
	Proper, mm	0 (−2.84, 9.865)	4.57 (0, 10.45)	0.02
	Deep, mm	34.19 (21.9, 46.18)	32.61 (23.02, 45.17)	0.99

CTD, the distance from the carina to tip of the CVC; CVC, central venous catheter; IQR, interquartile range.

**Table 2 jpm-12-01637-t002:** Outcomes of five-fold validation for classification of three labels according to the vertical distance from the carina to the CVC tip.

	1-Fold	2-Fold	3-Fold	4-Fold	5-Fold	Total
Overall Accuracy (F1 score)	0.76	0.74	0.71	0.80	0.76	0.75

**Table 3 jpm-12-01637-t003:** Confusion matrix (**A**) and outcomes (**B**) of performance test for classification of three labels of segmentation-free images after training with all segmented images.

**(A) Confusion Matrix**		**Labels**			
		**Shallow**	**Proper**	**Deep**	**Sum**
Prediction	Shallow	33	22	7	62
	Proper	16	127	30	173
	Deep	10	77	279	366
	Sum	59	226	316	601
**(B) Outcomes**		**Labels**			
		**Shallow**	**Proper**	**Deep**	**Average**
	Overall Accuracy	0.73			
	Accuracy	0.91	0.76	0.79	0.82
	Precision	0.53	0.73	0.76	0.73
	Recall	0.56	0.56	0.88	0.73
	F1 score	0.55	0.64	0.82	0.73

## Data Availability

The data presented in this study are available on request from the corresponding author.

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
