# Peer review of "Classification of Central Venous Catheter Tip Position on Chest X-ray Using Artificial Intelligence"

_jpm, 2022, doi:10.3390/jpm12101637_

Round 1
Reviewer 1 Report (Previous Reviewer 1)
The authors conducted a retrospective study to develop an algorithm for the classification of the central venous catheter tip position on chest radiography images. They labeled trachea and CVC on images, then used a two-stage approach model for automatic segmentation of the trachea and CVC with U-net++ and automatic classification of CVC placement with EfficientNet B4. Finally, they concluded that a deep CNN could achieve a comparative performance in the classification of the CVC position based on the distance from the carina to the CVC tip and automatic segmentation of the trachea and CVC on plain chest radiographs.
This is an interesting study with great value in clinical central venous catheterization. But further study is needed especially in bedside chest-X-ray.
Reviewer 2 Report (Previous Reviewer 4)
The authors have given reasonable answers to the questions and revised them in the manuscript.
This manuscript is a resubmission of an earlier submission. The following is a list of the peer review reports and author responses from that submission.
Round 1
Reviewer 1 Report
In this article, the authors developed an algorithm for the automatic classification of the proper depth based on the vertical distance between the tracheal carina and the CVC tip with the application of automatic segmentation of the trachea and the CVC on chest radiographs using a deep CNN. Here are some potential concerns.
1. The author mentioned some similar studies in background and discussion. What are the advantages and disadvantages of the author's algorithm compared with these studies?
2. How will this algorithm be applied in clinical practice in the future?
Reviewer 2 Report
This study is very weird, absolutely obsolete in its purpose, and it has no clinical interest in 2022.
In fact, all current guidelines (ESA 2020, INS 2021, etc.) recommend to verify the location of the tip during central venous catheterization by using non-invasive, accurate, intra-procedural methods such as intra-cavitary ECG or echocardiography. The chest-x-ray (even using the carina criteria) is a very inaccurate method for tip location (see INS 2021) and it has the major disadvantage of being post-procedural.
This article might have had some sense twenty years ago, but not now. We strongly invite the authors to update their clinical practice adopting appropriate methods of tip location such as intra-cavitary ECG (learn about it for example from this review: Pittiruti M, Pelagatti F, Pinelli F. Intracavitary electrocardiography for tip location during central venous catheterization: A narrative review of 70 years of clinical studies. J Vasc Access. 2021 Sep;22(5):778-785) or echocardiography (check these studies: Greca A, Iacobone E, Elisei D, Biasucci DG, D'Andrea V, Barone G, Zito Marinosci G, Pittiruti M. ECHOTIP: A structured protocol for ultrasound-based tip navigation and tip location during placement of central venous access devices in adult patients. J Vasc Access. 2021 Sep 8:11297298211044325. doi: 10.1177/11297298211044325. Epub ahead of print. Iacobone E, Elisei D, Gattari D, Carbone L, Capozzoli G. Transthoracic echocardiography as bedside technique to verify tip location of central venous catheters in patients with atrial arrhythmia. J Vasc Access. 2020 Nov;21(6):861-867. )
Reviewer 3 Report
General Comments
In their paper entitled “Automatic Segmentation and Three-Class Classification for Central Venous Catheter Position Based on the Distance from the Tracheal Carina to the Tip on Plain Chest X-ray Using Deep Convolutional Neural Networks”, authors describe the clinical use of AI software categorizing a CVC tip as shallow, proper or deep.
Specific Comments
Artificial Intelligence is gaining ground on imaging interpretation and a proper software for CVC identification and labeling would be welcomed in busy departments. Still, the human factor is not out of the equation yet, as overall accuracy of CVC positioning interpretation is lower than the human standard.
Title: The title may be too descriptive for a non-expert reader. I would suggest revising the title to something broader and shorter like: “Classification of Central Venous Catheter Tip Position on Chest X-ray Using Artificial Intelligence”
Abstract: No comment
Keywords: No comment
Introduction: No comment
1) The authors refer to a standard technique of inserting CVCs at a depth of 15 cm. This may be a good depth for the right IJV, but may lead to a shallow placement for a left IJV or a right or left SCV catheter. Please clarify.
2) It would be useful to mention the method of insertion of CVCs used, for example real-time ultrasound guided or landmark method or both, for IJV and SCV access.
3) For comparison purposes, we will have to assume all AP radiographs were taken using the same technique. For example “supine AP x-rays” and this should be clarified in the text.
4) Technical terms such as accuracy, recall and F1, may be hard to grasp for our readers. Authors are advised to elaborate more on the clinical implications of each number in the discussion section. For example what is the clinical implication of “the average accuracy, precision, recall and F1-score were 0.82, 0.73, and 0.73, respectively.”?
Reviewer 4 Report
The manuscript reports on an algorithm for automatic classification of proper depth based on the vertical distance between the tracheal carina and central venous catheter (CVC) tip with the application of automatic segmentation of the trachea and the CVC on chest radiographs using a deep CNN. However, I would recommend the authors to consider further information and revise the manuscript based on the following suggestions. My main comments are:
- there are some typos errors (e.g. in the Abstract " This study aimed to develop an algorithm for an automatic classification [...]" should be " This study aimed to develop an algorithm for automatic classification [...]"; in 2.6 Statistical Analysis, “Student’s t tests […]” should be “Student’s tests […]”). I recommend the authors to send the manuscript for proof reading.
- More information on the contribution of this article should be reported in abstract.
- What is the innovation of the proposed algorithm?
- More importantly, the main valuable results for this manuscript are shown in Table.1, Table.2 and Table.3, however, these results can not directly prove the effectiveness of the proposed algorithm.